# Physical Activity Intensity and Type 2 Diabetes: Isotemporal Substitution Models in the “Seguimiento Universidad de Navarra” (SUN) Cohort

**DOI:** 10.3390/jcm10132744

**Published:** 2021-06-22

**Authors:** María Llavero-Valero, Javier Escalada-San Martín, Miguel A. Martínez-González, Francisco Javier Basterra-Gortari, Alfredo Gea, Maira Bes-Rastrollo

**Affiliations:** 1Department of Preventive Medicine and Public Health, University of Navarra, 31008 Pamplona, Spain; mariallaverovalero@gmail.com (M.L.-V.); mamartinez@unav.es (M.A.M.-G.); javierbasterra@hotmail.com (F.J.B.-G.); ageas@unav.es (A.G.); 2Department of Endocrinology and Nutrition, University of Navarra, 31008 Pamplona, Spain; fescalada@unav.es; 3Biomedical Research Networking Center for Physiopathology of Obesity and Nutrition (CIBEROBN), Institute of Health Carlos III, 28029 Madrid, Spain; 4IDISNA, Navarra Institute for Health Research, 31008 Pamplona, Spain; 5Department of Nutrition, Harvard T.H. Chan School of Public Health, Boston, MA 02115, USA; 6Department of Endocrinology and Nutrition, Complejo Hospitalario de Navarra, 31008 Pamplona, Spain

**Keywords:** type 2 diabetes, strategies to prevent diabetes, vigorous physical activity, light to moderate physical activity, intensity, isotemporal substitution models, prospective

## Abstract

Which intensity of physical activity (PA) is associated with type 2 diabetes (T2D) prevention remains unclear. Isotemporal substitution models assess the relationship of replacing the amount of time spent in one activity for another. We aimed to assess T2D incidence associated with light-to-moderate physical activity (LMPA) and vigorous physical activity (VPA) using isotemporal substitution models of one hour (1 h) sitting by 1 h of LMPA or VPA. Furthermore, we evaluated the effect on T2D of an isotemporal substitution of 1 h sitting by 1 h of slow (light physical activity) or brisk–very brisk walking (moderate physical activity). In total, 20,060 participants (both sexes) of the SUN cohort (Spain) initially free of T2D followed-up during a median of 12 years were included. Cox regression models were fitted to assess the association between the substitution of 1 h LMPA, VPA, slow and brisk–very brisk pace by 1 h sitting and T2D. The replacement of 1 h sitting time by 1 h of VPA was associated with an adjusted HR of 0.52 (95% CI: 0.34–0.80), not observed for the substitution by 1 h of LMPA (HR 0.93; 95% CI: 0.73–1.20). An apparent inverse association was observed for the replacement of 1 h sitting time by 1 h of brisk/very brisk walking (HR: 0.69; 95% CI: 0.46–1.04), not observed by 1 h of slow pace. From equal conditions of duration and frequency of PA, the higher the intensity of PA, the greater the T2D prevention.

## 1. Introduction

Type 2 diabetes (T2D) has become a major public health issue across the globe. Despite the increased knowledge about healthy lifestyles in preventing this disease, the incidence of T2D continues to rise [1].

From a public health point of view, knowing how to spend our spare time and how it affects our health can be a helpful tool. In this context, isotemporal substitution models could be useful by estimating the effect of replacing one physical activity with another physical activity for the same amount of time [2,3]. There is a growing evidence indicating that time reallocation between sleep, sedentary behavior, light physical activity (LPA) and moderate-to-vigorous physical activity (MVPA) may be associated with mortality, general health, mental health, adiposity, fitness, and cardiometabolic biomarkers [4]. MVPA seems to be the most potent health-enhancing, time-dependent behavior, with additional benefit conferred from light-intensity activities and sleep duration when reallocated from sedentary time [5]. 

Regarding T2D, the American Diabetes Association (ADA) recommends increasing moderate intensity physical activity to at least 150 min/week. The ADA also encourages to break up prolonged sedentary time for T2D prevention [6]. Studies employing isotemporal models to evaluate the association of reallocating time from sedentary time into PA have demonstrated a pretty clear association between reallocating sedentary time to MVPA and improving T2D parameters [5,6,7,8,9], while the benefit of LPA is still controversial [10]. However, these studies have been conducted in populations affected by T2D or at high risk for it, including people with prediabetes. In addition, there is growing evidence suggesting an additional health benefit of vigorous physical activity (VPA) over moderate physical activity [11,12], probably due to greater significant reductions in fasting insulin, greater loss of overall and abdominal fat [13,14]. Given the lack of studies using isotemporal models to evaluate the association of the replacement of sedentary time by VPA, in our study, we assessed T2D incidence associated with VPA and light to moderate physical activity (LMPA) in an initially free T2D and prediabetes population using isotemporal substitution models. We also evaluated the association of LPA vs. a moderate one with T2D, defined as slow brisk and brisk–very brisk pace, respectively, through the replacement of 1 h sitting time by these both activities.

## 2. Materials and Methods

### 2.1. Study Population

The SUN project (Seguimiento Universidad de Navarra) is a prospective and multipurpose cohort with open enrolment. The recruitment started in 1999. SUN Project design details have been already published [15,16]. Its major aim is to evaluate the impact of lifestyle determinants and diet in the prevention of several diseases, like T2D, coronary heart disease, hypertension or depression. The cohort comprises Spanish university graduates of whom more than 50% are health professionals. Participants’ information is updated biennially by follow-up questionnaires. 

Up to December 2018, 22,790 participants had completed their baseline questionnaire. In order to assure at least 2 years of follow-up, we excluded 323 participants whose follow-up period was less than two years and nine additional months to account for the lag time in returning the questionnaire. We excluded 448 participants with total daily energy intake out of percentiles 1 and 99. We also excluded 475 participants not suitable for developing T2D (prevalent T2D, T1D and other type of diabetes, pancreatectomy, and with doubtful information in the additional confirmation questionnaire). Lastly, 1484 participants were lost to follow-up (retention: 93%). Finally, 20,060 participants were included in the analyses (Figure 1). The Institutional Review Board (IRB) of the University of Navarra approved the SUN project protocol. Voluntary completion of the first questionnaire was considered to imply informed consent.

### 2.2. Physical Activity and Sitting Time Assessment 

We assessed PA through a previously validated 17-item self-administered questionnaire [17]. This questionnaire requested data on the most popular free time activities: walking, jogging, athletics, cycling, stationary cycling, swimming, tennis, soccer, basketball, dance, hiking, gymnastics, gardening, skiing, martial arts and sailing. Frequency and time spent on each of these activities was also requested, taking into account seasonality variances. Each activity has its typical intensity expressed in activity metabolic equivalent (METs) [18]. We calculated the hours per day spent in each activity by dividing weekly hours per 7. We classified as light to moderate intensity (LMPA) those activities with less than 6 METs and vigorous activities (VPA) those with ≥6 METs. The following free time activities were considered as VPA: jogging, athletics, swimming, tennis, soccer, hiking, gymnastics, skiing, martial arts, and climbing stairs according to their METs [19]. The remaining activities were considered LMPA. Participants’ usual walking pace when walking down the street was also determined through the baseline questionnaire. It included the following four possible close-ended options for the response: “slow”, “normal–moderate”, “brisk”, “very brisk”. Self-reported walking speed has been proven to be a good marker of measured walking speed [20] and it has been used in several studies [21,22]. Metabolic energy equivalents (METS) were assigned to each pace—slow: 2.5 METS, normal/moderate 3.16 METS, brisk: 3.8 METS, very brisk: 4.5 METS. After that, we used the Compendium of Physical Activity [19], to calculate the MET score for each pace by considering the self-reported pace of slow as 2.0 mph (3.2 km/h), average as 2.5–3.2 mph (4.0–5.1 km/h), brisk as 3.5 mph (5.6 km/h), and very brisk as 4.0 mph (6.4 km/h).

Information on sitting time was also gathered at baseline questionnaire inquiring about the total hours per day that a participant spends sitting during a typical day during the week and during the weekend. We calculated a weighted mean to obtain the average total hours of sitting per day. 

### 2.3. Outcome Assessment 

Participants who, at baseline questionnaire, reported T2D or were under treatment (insulin or oral antidiabetic drugs) were excluded. Participants were asked for new onset T2D in each of the follow-up questionnaires. If a T2D diagnosis was reported, an additional questionnaire was sent requesting further information as well as the medical report detailing different aspects relevant for the diagnosis of T2D (confirmation of diagnosis, type of diabetes, date of diagnosis, highest glucose level objectively measured, glycated haemoglobin, oral glucose tolerance test if it had been performed and glucose-lowering medications). An endocrinologist blinded to the exposure adjudicated incident cases, according to the American Diabetes Association criteria [6]. 

### 2.4. Other Covariates

Anthropometrics, sociodemographic characteristics, lifestyle habits, and medical history were inquired in the baseline questionnaire. Information on dietary intakes was collected by a repeatedly validated self-administered 136-item semiquantitative food frequency questionnaire (FFQ), used in Spain [23,24,25]. We categorized the adherence to the Mediterranean dietary according to Trichopoulou’s et al. 9-point score [26].

### 2.5. Statistical Analyses

Baseline participants’ characteristics across quartiles of VPA and sitting time were adjusted for age and sex using the inverse probability method. 

We firstly conducted a Cox regression analysis to study the association between LMPA or VPA and incident T2D.

For each participant, we computed person-years of follow-up from the date of the last follow-up questionnaire, the date of T2D diagnosis, or the date of death, whichever occurred first.

We performed Cox regression analyses to assess the association between incident T2D and the replacement of one hour sitting by one hour of LMPA or VPA. Hazard ratios (HR) and 95% confidence intervals for isotemporal substitutions were estimated as the difference between β coefficients of the two activities studied and then exponentiated. The HR reflects the reduction in T2D incidence that is observed when the mean time spent in vigorous physical activity increased by 1 h/day because the mean time spent in sitting time is decreased by 1 h/day.

In the Cox regression models, age was considered as the underlying time scale. We performed an age- and sex-adjusted model, and a multivariable model additionally adjusted for smoking status, family history of diabetes, prevalent hypertension, cancer or cardiovascular disease at baseline, adherence to the Mediterranean dietary pattern according to Trichopoulou’s 9-point score, soft drink consumption, snacking between meals and overweight/obesity at baseline. All Cox models were stratified by categories of age (decades) and the date of entry into the cohort (3 categories). 

We also conducted sensitivity analyses by rerunning all the models under different a priori established assumptions: including only women, including only men, excluding participants with family history of T2D, excluding late cases of T2D (those occurring >=10 years of follow-up), applying Willett’s energy limits or percentiles 5–95 of total energy intake, and excluding participants with prevalent hypertension, cancer, cardiovascular disease or hypercholesterolemia.

A potential interaction by amount of physical activity on the association between the replacement of 1 h sitting time by 1 h vigorous physical activity and T2D, was also assessed using the likelihood ratio test stratifying participants based on the median value of physical activity. 

Finally, we compared the association of a light physical activity (<3 METs) vs. a moderate one (≥3–5.9 METs) with T2D; for this purpose, we examined the relationship between walking pace and T2D, considering slow pace as LPA and brisk-very brisk pace as MPA. An isotemporal replacement of one hour sitting by one hour of slow or brisk–very brisk was performed. 

All *p*-values < 0.05 were considered as statistically significant. Analyses were performed using STATA/SE V.12.1 (StataCorp, College Station, TX, USA).

## 3. Results

Our study included 20,060 participants with a median follow-up of 12 years (61.5% women; mean age 37.4 ± 12.2 years). Baseline characteristics across quartiles of VPA and sitting time are represented in Table 1. 

When comparing participants between the lowest and the highest category of VPA, those at the highest category had less prevalent hypertension and cardiovascular disease at baseline. They had a better adherence to Mediterranean diet, and they tended to snack between meals less frequently compared to those at the lowest quartile of VPA. They also spent less time watching TV compared to participants in the lowest quartile of VPA. As expected, they had a lower BMI. Regarding sitting time classification, participants at the highest rate of sitting time had more prevalent depression and hypercholesterolemia compared with participants who spent less time sitting. They also were more likely to be current smokers and had higher education level. On average, they spent more hours on TV viewing, had a higher BMI and had a worse adherence to Mediterranean diet

During the follow-up, we identified 175 incident cases of T2D (T2D incidence 0.87%). In the Cox regression analyses assessing the association of LMPA or VPA and T2D, a relative risk reduction of 12% was found for each hour per day doing LMPA although it was not statistically significant (HR 0.88 (95% CI 0.69–1.12); *p* = 0.291), while a significant 47% relative reduction in the risk of T2D (HR 0.53 (95% CI 0.35–0.80); *p* = 0.003) was found for each additional hour per day doing VPA, after adjusting for multiple confounders. 

The replacement of one hour sitting time by one hour of VPA was associated with a HR of 0.52 (95% CI, 0.34–0.80; *p* = 0.003) in the multivariable-adjusted model. Non-statistically significant differences were seen for the replacement of one hour sitting time by one hour of LMPA (Table 2). 

Sensitivity analyses were conducted. No substantial changes were observed (Table 3).

Interaction analyses on the association between the replacement of 1 h sitting time by 1 h VPA and T2D, by amount of physical activity, approached statistical significance (*p* for interaction = 0.09). Therefore, although the results were not statistically significant at the conventional alpha level of 5%, those participants with lower physical activity, exhibited a stronger inverse association with T2D development by the replacement of 1 h sitting by 1 h of VPA (adjusted HR = 0.14; 95% CI: 0.02–1.30; *p* = 0.08) as compared to the corresponding association observed among those with higher levels of physical activity (adjusted HR = 0.40; 95% CI: 0.23–0.68; *p* = 0.001).

Finally, in order to study the separate effect of light PA (LPA) vs. moderate PA (MPA) since it was englobed into the LMPA category, we analyzed the effect of walking pace through the replacement of one hour sitting time by one hour of slow (LPA) or quick to very quick pace (MPA). An inverse association with the risk of T2D was observed for the replacement of 1 h sitting time by 1 h of brisk or very brisk walking, although it only approached statistical significance (adjusted HR: 0.69 (95% CI: 0.46–1.04); *p* = 0.076). On the other hand, the replacement of 1 h sitting time by 1 h of slow pace did not support any protection against T2D (adjusted HR 1.04 95% CI (0.73–1.47), *p* = 0.844) (Figure 2).

## 4. Discussion

The incidence of T2D in our Mediterranean cohort was very low (0.87%), probably mainly due to the young age of the participants, their low body mass index, their relatively good adherence to the Mediterranean diet [27], their high educational level and the high prevalence of leisure-time physical activity. Notwithstanding, even in this situation of a low absolute risk, VPA was associated with a significant relative risk reduction of 48% of T2D. However, no significant reduction was observed for LMPA.

The results of isotemporal substitution models are in line with these results, since only the replacement of one hour sitting time by one hour of VPA, achieved a statically significant risk reduction in T2D incidence. On the other hand, when we assessed the effect of LPA and moderate physical activity (MPA) separately, through the impact of walking pace by the replacement of one hour sitting by one hour low pace (LPA) walking or by one hour brisk/very brisk pace (MPA) walking, only the moderate intensity activity showed an inverse association in terms of T2D protection, although its statistical significance was not reached. Therefore, under equal conditions of duration and frequency, the intensity level of PA is a key factor in T2D prevention. The non-statistically significant results observed in our study in the replacement of one hour sitting by one hour of LMPA associated with T2D prevention, could be due to a strong contribution of the light physical activity. 

The reallocation of sedentary time to MVPA has widely demonstrated its cardiometabolic benefits, including improvement of T2D parameters [10,12,28]. However, the benefits of the replacement of sedentary time by LPA for T2D prevention are not so well established, although most of the studies point to its protective effect [9,29,30]. The PREDIMED- Plus study, which is conducted in a Spanish population too, demonstrated in 2019 that isotemporal substitution of inactive time with MVPA and LPA could have beneficial impacts on cardio-metabolic health, including T2D [31]. Nonetheless, Rossen et al. [10] demonstrated that reallocation of sedentary time in bouts as well as non-bouts to MVPA, was beneficially associated with waist circumference, BMI and HDL cholesterol in individuals with prediabetes and type 2 diabetes, but not to LPA. Most of these studies are conducted in older populations, with established diabetes or at high risk. It should be noted that the SUN cohort mean age is 37.4 ± 12.2 years. In this respect, the CARDIA Study has recently demonstrated that replacing sedentary time or LPA with MVPA had consistent beneficial associations with cardiovascular risk in an early adulthood cohort too (mean age 45.2 ± 3.6 years) [32]. In our study, we examined the reallocation of one hour sitting time to two different categories of PA intensity in a young cohort and non-obese. Of note, our population is also not at risk of T2D. Only the reallocation of one hour sitting time by one hour of VPA showed a reduction of T2D development when compared with LMPA. Our results showed that the higher intensity, the greater prevention in T2D. These findings were consistent with previous evidence suggesting additional benefit of greater proportion of MVPA volume performed at a vigorous intensity in all-cause mortality rates [33] and risk of major chronic disease especially in men [34]. The SUN cohort had also previously showed the benefit of VPA over LMPA in preventing metabolic syndrome, independent of total time and total energy spent in leisure time physical activity [12] and we had also found a lower risk of cardiovascular disease given the same level of energy expenditure in those participants engaged in higher intensity leisure time physical activity [12]. 

On the other hand, interaction analyses showed that those participants with lower physical activity exhibited a higher protection against T2D development with the substitution of 1 h sitting for 1 h of VPA compared to those with less physical activity. Nonetheless, the p for interaction only approached the statistical significance in our study. This finding is in line with the latest World Health Organization (WHO) guidelines on physical activity and sedentary behaviors [35]. In order to minimize the negative impact of sedentariness, WHO guidelines support the recommendation of reducing time in sedentary behaviors or increasing MVPA, or some combination of both strategies.

From a pathophysiological point of view, MVPA had demonstrated a negative association with adiposity, that had not been observed in sedentary breaks [36]. Jelleyman et al. suggested that the higher the intensity, the greater the potential improvement for a given PA duration in regard to insulin sensitivity assessed by HOMA-IS and Matsuda index [37]. Likewise, Henson et al. highlighted the convenience of more intense PA, as they found a higher reduction in chronic inflammation markers (C-reactive protein, interleukin 6) with higher intensity PA [7].

The rapidly increased prevalence in T2D requires effective measures. The American Diabetes Association recommends at least 150 min/week of physical activity of intensity at least similar to brisk walking. Our findings are in line with this guideline. Moreover, according to our results, the higher intensity the greater benefit in terms of T2D prevention. VPA should be encouraged especially in the young adult population. Nevertheless, there is increasing evidence suggesting the importance of an individualized PA prescription [38,39].

### Strengths and Study Limitations

There are several strengths of our study including its large sample size with a long follow-up, a high retention rate (93%) and the use of previously published validation studies. The present study has also some limitations. First, the number of observed cases of T2D was not large and some non-statistically significant findings could be derived from a suboptimal statistical power. Second, it is an observational study and, therefore, residual confounding cannot be totally excluded. However, we adjusted the multivariate analysis for a wide number of confounders. Additionally, we assessed PA intensity through a validated questionnaire, but some misclassification of exposure is inevitable. Some participants may perform activities that we would classify as vigorous at a current intensity that is lower or vice versa. Nevertheless, the cohort comprises university graduates, of which more than 50% are health professionals. This fact reduces the representativeness of the sample and we should be cautious regarding external validity but, on the other hand, the high educational level of the cohort adds validity to the self-reported information derived from their questionnaires and also provides a homogenous study sample with less room for residual confounding by socioeconomic factors, given that restriction is an excellent technique in epidemiology to reduce confounding. Third, the physical activity questionnaire does not capture all the possible movement-related behaviors and therefore they were not taken into account for the analyses. Finally, the present study is based on mathematical models, and does not correspond to an intervention trial. However, it helps to decide whether an interventional study would be worthwhile.

## 5. Conclusions

Our cohort provides additional evidence that the higher the intensity of PA, the greater the benefit in T2D reduction, at least in a young Mediterranean population. It seems that the replacement of sedentary behavior by LPA does not lead to an improvement in TD2 prevention at least in our population study. From a public health perspective, the recommendation of walking quickly or very quickly for one hour instead of one hour sitting was associated with a substantial reduction in the risk of T2D. This simple message could be a useful tool to empower patients to take care of their health and improve primary prevention of T2D.

## Figures and Tables

**Figure 1 jcm-10-02744-f001:**
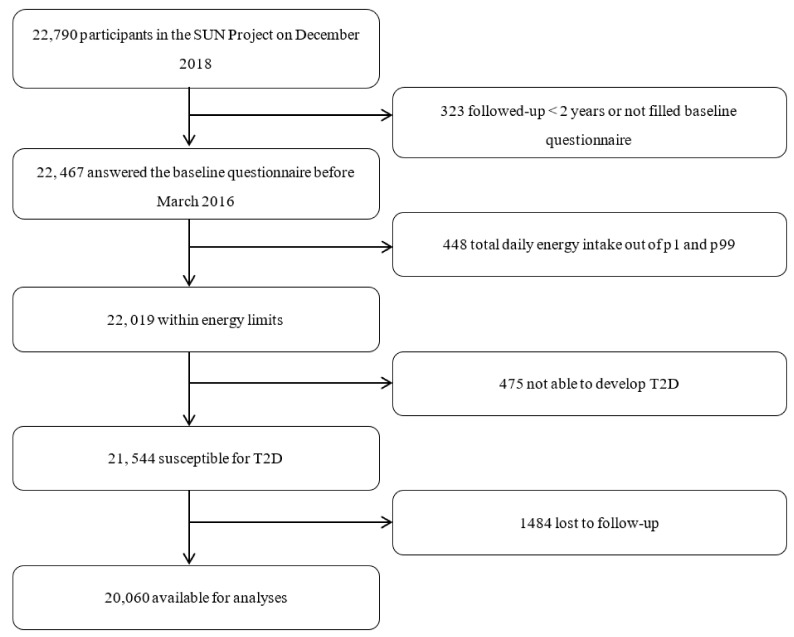
Flow chart of participants.

**Figure 2 jcm-10-02744-f002:**
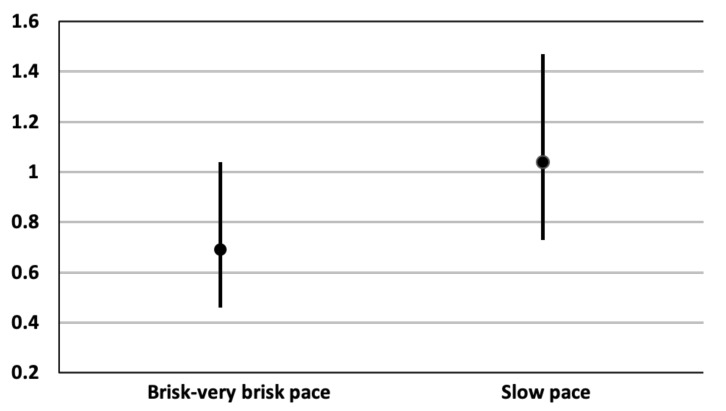
Adjusted * hazard ratios for T2D of the replacement of 1 h sitting time by 1 h of walking according to pace. * Adjusted for sex, soft drinks consumption (tertiles), smoking, family history of diabetes, prevalent hypertension, cancer, cardiovascular disease, snacking, Mediterranean diet score, and baseline overweight/obesity, stratified by decades of age and recruitment period. Additionally, adjusted for MPA and VPA in the replacement of 1 h sitting time by 1 h of slow pace model and for LPA and VPA in the replacement of 1 h sitting time by 1 h of the brisk/very brisk pace model.

**Table 1 jcm-10-02744-t001:** Age- and sex-adjusted * characteristics of the SUN participants according to quartiles of sitting time and VPA.

	Quartiles of VPA According to METS					Quartiles of Sitting Time				
	VPA1	VPA2	VPA 3	VPA 4	*p*-Value **	SIT1	SIT2	SIT3	SIT4	*p*–Value **
**N**	5009	5010	5016	5023		5556	4520	5084	4900	
Physical activity (METs-h/wk)	7.3 (3.1)	22.1 (4.8)	43.3 (7.9)	100.3 (42.8)	<0.001	45.2 (44.6)	49.0 (47.2)	42.6 (39.9)	38.9 (39.2)	<0.001
Age (years)	37.4 (11.8)	37.4 (12.2)	37.4 (12.3)	37.4 (12.3)	0.9	37.3 (11.7)	37.4(12.3)	37.4 (12.4)	37.4 (12.5)	<0.001
Women (%)	61.6	61.7	61.5	61.7	0.79	60.0	62.0	61.4	61.0	<0.001
BMI (kg/m^2^)	23.9 (3.8)	23.5 (3.6)	23.4 (3.4)	23.0 (3.1)	<0.001	23.4 (3.4)	23.3 (3.4)	23.5 (3.5)	23.6 (3.7)	<0.001
Smoking status (%)					<0.001					<0.001
Never	44.6	48.1	48.9	51.2		48.2	48.9	49.2	47.6	
Current	26.2	22.8	21.6	18.4		21.8	22.2	21.4	24.1	
Former	28.2	28.1	28.7	29.6		30.0	28.9	29.5	28.4	
Education level (%)					<0.001					<0.001
Graduate	73.6	72.4	72.4	71.1		79.5	73.0	68.9	6.1	
Postgraduate	8.3	7.1	8.2	8.5		5.8	6.2	9.1	10.8	
Doctorate	9.2	10.5	9.90	10.1		7.0	8.1	12.9	13.6	
Family history of T2D (%)	15.7	15.1	14.7	15.0	0.21	16.5	14.4	14.3	15.6	<0.001
Prevalent hypertension (%)	19.5	19.6	18.4	17.6	<0.001	19.2	18.7	18.8	19.2	0.31
Prevalent cancer (%)	3.0	2.9	3.6	3.3	0.19	3.1	2.8	3.4	3.9	0.21
CVD at baseline (%)	1.3	1.5	1.4	1.6	0.03	1.3	1.4	1.5	1.7	0.90
Hypercholesterolemia (%)	16.7	16.9	17.1	15.9	0.07	16.5	15.0	17.2	18.0	<0.001
Prevalent depression (%)	12.9	11.1	11.3	10.6	0.11	10.6	11.5	11.0	12.7	0.02
Mediterranean dietary pattern (0–9 points) ^†^	3.9 (1.8)	4.1 (1.8)	4.3 (1.8)	4.6 (1.8)	<0.001	4.3 (1.8)	4.3 (1.8)	4.2 (1.8)	4.1 (1.8)	<0.001
Total energy intake (kcal/day)	2452.8 (783.0)	2467.2 (764.8)	2514.8 (759.9)	2583.1 (795.4)	<0.001	2529.1 (791.2)	2515.6 (774.4)	2501.3 (772.9)	2489.6 (777.9)	0.89
Snacking (%)	38.1	35.3	34.6	31.4	<0.001	35.4	35.8	34.2	34.7	0.41
Soft drinks (portions/day)	0. 2 (0.5)	0.2 (0.5)	0.2 (0.4)	0.2 (0.4)	<0.001	0.2 (0.5)	0.2 (0.4)	0.2 (0.4)	0.2 (0.4)	<0.001
TV watching (hours)	1.7 (1.2)	1.7 (1.2)	1.6 (1.2)	1.6 (1.1)	<0.001	1.5(1.2)	1.7 (1.1)	1.7(1.2)	1.8 (1.2)	<0.001

Continuous variables are expressed as means and (standard deviations) and categorical variables as percentages. * Adjusted through inverse probability weighting, ** through ANOVA and Chi-squared test weighted by the inverse probability weighting method ^†^ Assessed with Trichopoulou’s 9-point score [20], PA: physical activity, SIT: sitting time, MET: activity metabolic equivalent, BMI: body mass index, CVD: cardiovascular disease.

**Table 2 jcm-10-02744-t002:** HR for incident T2D associated with the substitution of one hour of sitting for one hour of vigorous or light to moderate physical activity.

	Vigorous Physical Activity			Light to Moderate Physical Activity		
	HR	95% CI	*p*-Value	HR	95% CI	*p*-Value
Age and sex adjusted	0.41	0.27–0.64	<0.001	0.94	0.74–1.21	0.65
Multivariable adjusted *	0.52	0.34–0.80	0.003	0.93	0.73–1.20	0.59

* Adjusted for age (underlying time variable), sex, soft drinks consumption, prevalent hypertension, smoking status, family history of T2D, Mediterranean diet score, baseline overweight/obesity, prevalent cancer, prevalent cardiovascular disease, snacking and vigorous or light-to-moderate physical activity depending on the analyses, stratified by decades of age and recruitment period.

**Table 3 jcm-10-02744-t003:** Sensitivity analysis for incident T2D associated with the substitution of one hour of sitting for one hour of vigorous physical activity.

Variable	N	Incident T2D	HR (95% CI)
Overall *	20,060	175	0.52 (0.34–0.80)
Including only women	12,344	43	0.66 (0.27–1.61)
Including only men	7716	132	0.49 (0.30–0.80)
Excluding family history of T2D	17,029	107	0.35 (0.19–0.66)
Excluding late cases of T2D (≥10 first years)	20,030	145	0.56 (0.36–0.88)
Excluding prediabetes	20,044	175	0.52 (0.34–0.80)
Willett’s energy limits	18,307	166	0.53 (0.34–0.81)
Energy limits: percentiles 5–95	18,056	159	0.53 (0.34–0.83)
Excluding prevalent hypertension	16,286	66	0.49 (0.24–0.98)
Excluding prevalent cancer	19,414	164	0.50 (0.32–0.78)
Excluding prevalent cardiovascular disease	19,776	163	0.51 (0.33–0.79)
Excluding prevalent hypercholesterolemia	19,432	170	0.53 (0.35–0.82)

* Adjusted for age (underlying time variable), sex, soft drinks consumption, prevalent hypertension, smoking status, family history of T2D, Mediterranean diet score, baseline overweight/obesity, prevalent cancer, prevalent cardiovascular disease, snacking and light-to-moderate physical activity, stratified by decades of age and recruitment period.

## Data Availability

The data presented in this study are available on request from the corresponding author. The data are not publicly available.

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
