# Peer review of "Physical Activity Intensity and Type 2 Diabetes: Isotemporal Substitution Models in the “Seguimiento Universidad de Navarra” (SUN) Cohort"

_jcm, 2021, doi:10.3390/jcm10132744_

Round 1

Reviewer 1 Report

The aim of this manuscript was to assess how substituting sitting time with VPA, MPA, and LPA differentially effect the incidence of T2D in younger adults who are not at risk for T2D. Although the isotemporal substitution of sedentary behaviors with other intensities of PA is not new, this study utilizes a population that hasn’t previously been widely looked at, which is a younger population with no known pre-disposition for T2D. I can see that the assessment of the effect of PA in a young healthy population without any other pre-existing risk of T2D can provide some potential value and insight as the use of this population may better help isolate the independent role of PA on the development of the T2D. However, I think the paper can be further improved.

My major comment is on the English. The manuscript has multiple grammatical errors making it difficult to read.

For instance:

-Line 35-26-“ Despite the largely knowledge about healthy lifestyles in preventing this disease, T2D incidence continues raising” should read “Despite the increased knowledge about healthy lifestyles in preventing this disease, the incidence of T2D continues to rise.”

-Line 48-50 “Regarding T2D, the American Diabetes Association recommends increasing moderate intensity physical activity to at least 150 minutes/week as well as it encourages to break up prolonged sedentary time for its prevention.” “it” is used multiple times in this sentence and the reader loses track to what “it” is referring to.

-Line 54 “….these studies have conducted in populations..” should read “….these studies have been conducted in populations..”

-Line 99 “the total hours per day that a participant is sit during a typical day” should read “the total hours per day that a participant spends sitting during a typical day”

Other comments:

Methods

-Lines 76-79 “We also excluded 475 participants not suitable for developing T2D (prevalent T2D, T1D and other type of diabetes, pancreatectomy, and with doubtful information in the additional confirmation questionnaire).” What did you define as “not suitable for developing T2D”?

-How was slow pace and brisk to very brisk pace determined by the questionnaire? Is it basically just LPA and MPA?

Results

-In the second paragraph of the results, it states that there were differences between PA categories and sitting time categories for certain variables. It would be helpful if these differences are indicated in Table 1 with and asterisk.

-In Table 2 and 3, for the models where sitting time was substituted with VPA, I would suggest adjusting for light-to-moderate and not just moderate PA.

-In Table 3, are you substituting VPA with sitting time or sitting time with VPA? I am assuming sitting time with VPA, but the title of Table 3 states otherwise.

-Figure 2: What happens if these models are adjusted by VPA? Also, what about adjusting for MPA in models with light walking and LPA in models with brisk walking? In order to fully assess the effect of replacing sitting with these behaviors, I believe you need to hold all the other activities constant.

-It would be interesting to assess if replacing sitting time with VPA or MPA (brisk walking) had a different effect on T2D incidence in people who were already in the highest quartile of PA vs. those in the lowest. An effect modification between sedentary time and MVPA has been acknowledged and highlighted recently in the new WHO PA guidelines (doi: 10.1136/bjsports-2020-102955). Thus, I think it is relevant to investigate this.

Discussion

-In limitations section I think it is important to mention that no all movement-related behaviours (e.g., sedentary behaviour, sleep, LPA, MPA, and VPA) were included.

Author Response

Major comment

Point 1: My major comment is on the English. The manuscript has multiple grammatical errors making it difficult to read.

For instance:

-Line 35-36- “Despite the largely knowledge about healthy lifestyles in preventing this disease, T2D incidence continues raising” should read “Despite the increased knowledge about healthy lifestyles in preventing this disease, the incidence of T2D continues to rise.”

-Line 48-50 “Regarding T2D, the American Diabetes Association recommends increasing moderate intensity physical activity to at least 150 minutes/week as well as it encourages to break up prolonged sedentary time for its prevention.” “it” is used multiple times in this sentence and the reader loses track to what “it” is referring to.

-Line 54 “…. these studies have conducted in populations...” should read “…these studies have been conducted in populations...”

-Line 99 “the total hours per day that a participant is sit during a typical day” should read “the total hours per day that a participant spends sitting during a typical day”

Response 1. We thank you for this comment. We have carefully reviewed the English grammar and style and have corrected these sentences than you mentioned. We have also reworded some other sentences. A native English speaker has also revised the new version of the manuscript before this resubmission.

Other comments:

Methods

Point 1: Lines 76-79 “We also excluded 475 participants not suitable for developing T2D (prevalent T2D, T1D and other type of diabetes, pancreatectomy, and with doubtful information in the additional confirmation questionnaire).” What did you define as “not suitable for developing T2D”?

Response 1: As we mentioned (in brackets), we excluded those participants who had been already diagnosed of T2D at baseline, participants with T1D or other specific types of diabetes such as diabetes secondary to other causes, including diseases of the exocrine pancreas or pancreatectomy, as well as those participants who provided unclear information when responding to the additional specific questionnaire for T2D confirmation.

Point 2: How was slow pace and brisk to very brisk pace determined by the questionnaire? Is it basically just LPA and MPA?

Response 2: The intensity of the walking pace was a specific item in our baseline questionnaire. This item in the baseline questionnaire inquired about participants’ usual walking pace and included the following four possible close-ended options for the response: “slow”, “normal-medium”, “brisk”, “very brisk”.  According to the 2011 Compendium of Physical Activities (Ainsworth BE, et al. Med Sci Sport Exerc. 2011; 43 (8): 1575-1581), walking intensity for each pace ranges from 2.5 for slow pace to 4.5 METS for very brisk pace. Therefore, we agree that walking should be considered as LPA or MPA, not VPA.

In this context, it is noteworthy that we recently reported that walking pace (measured as we mention above) was found to be independently and inversely associated with the incidence of hypertension in our cohort (Etzig C, Gea A, Martínez-González MÁ, Sullivan MF Jr, Sullivan E, Bes-Rastrollo M. The association between self-perceived walking pace with the incidence of hypertension: the 'Seguimiento Universidad de Navarra' cohort. J Hypertens. 2021;39:1188-1194).  

Results

Point 1: In the second paragraph of the results, it states that there were differences between PA categories and sitting time categories for certain variables. It would be helpful if these differences are indicated in Table 1 with and asterisk.

Response 1: The differences pointed out in that paragraph referred to Table 1 which showed participants’ age and sex-adjusted baseline characteristics according to quartiles of PA and sitting time.  Therefore, we reported means with their standard deviations for quantitative variables and percentages (%) for categorical variables. All variables were adjusted for age and sex using inverse probability weighting.

Nonetheless, for the revised manuscript, we have now calculated p values from ANOVA and Chi-squared test corrected for sex and age with the use of the inverse probability weighting method. We have also rewritten the manuscript accordingly.

Point 2: In Table 2 and 3, for the models where sitting time was substituted with VPA, I would suggest adjusting for light-to-moderate and not just moderate PA.

Response 2: Thank you for pointing this out. Indeed, it was a typo. Both analyses were adjusted for light-to-moderate physical activity.

Point 3: In Table 3, are you substituting VPA with sitting time or sitting time with VPA? I am assuming sitting time with VPA, but the title of Table 3 states otherwise.

Response 3: Yes, we apologize for the misunderstanding. We have rewritten the title. Thank you for your appreciation.

Point 4: Figure 2: What happens if these models are adjusted for VPA? Also, what about adjusting for MPA in models with light walking and LPA in models with brisk walking? In order to fully assess the effect of replacing sitting with these behaviors, I believe you need to hold all the other activities constant.

Response 4: According to the reviewer’s comment we have adjusted both models for VPA as well for LPA and MPA, respectively. After these adjustments, an inverse association with the risk of T2D was observed for the replacement of one hour sitting time by one hour of brisk or very brisk walking, although it only approached statistical significance [adjusted HR: 0.69 (95% CI: 0.46-1.04); p=0.076]. On the other hand, the replacement of one hour sitting time by one hour of slow pace did not support any protection against T2D [adjusted HR 1.04 95% CI (0.73-1.47), P=0.844].

We have rewritten the new revised manuscript with these results.

Point 5: It would be interesting to assess if replacing sitting time with VPA or MPA (brisk walking) had a different effect on T2D incidence in people who were already in the highest quartile of PA vs. those in the lowest. An effect modification between sedentary time and MVPA has been acknowledged and highlighted recently in the new WHO PA guidelines (doi: 10.1136/bjsports-2020-102955). Thus, I think it is relevant to investigate this.

Response 5: You raised an interesting point. For the new version of the manuscript, we have assessed a potential interaction by amount of physical activity on the association between the replacement of 1 h-sitting time for 1 h-vigorous physical activity and T2D. The multiplicative interaction-term only approached statistical significance (p for interaction=0.09).  So, although the results were not statistically significant at the conventional alpha level of 5%, those participants with lower physical activity, exhibited a stronger inverse association with T2D development by the replacement of 1-h sitting by 1-h of VPA (adjusted HR=0.14; 95% CI: 0.02-1.30; p=0.08) as compared to the corresponding association observed among those with higher levels of physical activity (adjusted HR=0.40; 95% CI: 0.23-0.68; p=0.001). We have discussed this result in the new version of the manuscript considering also the latest WHO PA guidelines, as you suggested. 

Discussion

Point 1: In limitations section I think it is important to mention that not all movement-related behaviours (e.g., sedentary behaviour, sleep, LPA, MPA, and VPA) were included.

Response 1: We completely agree with the reviewer´s suggestion. Accordingly, we have mentioned the limitation that not all movement-related behaviours were included in our assessments.

Reviewer 2 Report

The present work addresses an important area of concern for assessment of Type 2 diabetes ( T2D) incidence associated with vigorous physical activity (VPA) and light to moderate physical activity (LMPA) in an initially free T2D and prediabetes population using isotemporal substitution models. In addition, the authors evaluated the association of LPA vs a moderate PA with T2D through the replacement of 1 hour sitting time by these both activities. Considering the large sample size with a long follow-up, this well-designed work is of great clinical relevance. The manuscript is well conducted.

Please find here below some remarks and suggestions to still enhance the quality of this paper.

MINOR COMMENTS

  • Abstract

If possible, in the abstract, indicate that subjects were both males and females.

In the conclusion of the abstract, you should maybe explain, in a few words, what is "isotemporal substitution".

  • Introduction

Line 39-40 : please add a reference after this sentence “In this context, isotemporal substitution models .... for the same amount of time”

Line 55- 56 : Please explain from a physiological point of view why VPA provide an additional health benefit  comparing to  moderate physical activity

  • Material and methods

Please adjust the following words in figure 1: questionnaire, March 2016, 2018

  • Statistical Analysis

It should be reported what test was used to test the normality of the data and what test were used to test the significance of differences.

  • Results

Table 1 : Please add the meaning of  PA and SIT below the table.

How authors can conclude that participants at the highest category of PA had less prevalent hypertension and depression?  Has a statistical test been done?

  • Discussion

More discussion on existing studies ( e.g DOI: 10.1186/s12966-019-0892-4 ;   DOI: 10.1016/j.ypmed.2021.106626  …) using isotemporal  substitution models in the context of type 2 diabetes prevention is recommended.

Author Response

Abstract

Point 1: If possible, in the abstract, indicate that subjects were both males and females.

Response 1: OK. Done.

Point 2: In the conclusion of the abstract, you should maybe explain, in a few words, what is "isotemporal substitution".

Response 2:  Done. Thank you.

Introduction

Point 1: Line 39-40: please add a reference after this sentence “In this context, isotemporal substitution models .... for the same amount of time”

Response 1: To comply with your kind suggestion, we have added two references  after that sentence.

Point 2: Line 55- 56: Please explain from a physiological point of view why VPA provides an additional health benefit comparing to moderate physical activity.

Response 2: As you suggested, we have further discussed the additional health benefits of VPA as compared to moderate physical activity in the new version of the manuscript.

Material and methods

Point 1: Please adjust the following words in figure 1: questionnaire, March 2016, 2018

Response 1: Done. Thank you for pointing this out.

Statistical Analysis

Point 1: It should be reported what test was used to test the normality of the data and what test were used to test the significance of differences.

Response 1: We thank you again for your comment. We did not formally tested the normality of the data due to the large population included for the analysis (n=20,060 participants). Please check https://pubmed.ncbi.nlm.nih.gov/11910059/.

In regard to the analyses performed to test the significance of differences, as we explained in the methods sections, we conducted Cox regression models. Hazard ratios (HR) and 95% confidence intervals for isotemporal substitutions were estimated as the difference between b coefficients of the two activities studied and then exponentiated. The inclusion or exclusion of the null value (HR=1.00) in the width of the 95% confidence intervals for the HRs already includes the relevant information about testing the null hypothesis (please check Amrhein et al. Nature. 2019;567:305-7, and Wasserstein RL, Lazar NA. The ASA's Statement on p-Values. Am Stat 2018 (https://www.tandfonline.com/doi/full/10.1080/00031305.2016.1154108). The HR reflects the reduction in T2D incidence that is observed when the mean time spent in LMPA/VPA increased by 1 hour/day because the mean time spent in sitting time is decreased by 1 hour/day. 

Results

Point 1: Table 1: Please add the meaning of PA and SIT below the table.

Response 1: Done. Thank you

Point 2: How authors can conclude that participants at the highest category of PA had less prevalent hypertension and depression?  Has a statistical test been done?

Response 2: We thank you for your kind comment. In the original manuscript, we only presented a description of baseline characteristics. Table 1 showed participants’ age- and sex-adjusted baseline characteristics according to quartiles of VPA and sitting time. Therefore, we reported means with their standard deviations for quantitative variables and percentages (%) for categorical variables. All variables were adjusted for age and sex using inverse probability weighting.

Nonetheless, to comply with your comment, we have now calculated p values obtained using ANOVA and Chi-squared tests, weighted by the inverse probability weighting method. We have also rewritten the manuscript accordingly.

Discussion

Point 1: More discussion on existing studies (e.g DOI: 10.1186/s12966-019-0892-4; DOI: 10.1016/j.ypmed.2021.106626 …) using isotemporal substitution models in the context of type 2 diabetes prevention is recommended.

Response 1: Thank you again. To comply with your suggestion, we now mention and have included references to additional studies supporting our study hypothesis.